# Application of the Hill-Wheeler Formula in Statistical Models of Nuclear Fission: A Statistical–Mechanical Approach Based on Similarities with Semiconductor Physics

**DOI:** 10.3390/e27030227

**Published:** 2025-02-22

**Authors:** Hirokazu Maruyama

**Affiliations:** Independent Researcher, Kobe 655-0861, Hyogo, Japan; etctransformation@jcom.zaq.ne.jp

**Keywords:** nuclear fission, statistical model, Hill–Wheeler equation, quantum mechanical distribution function, distribution function, fission yield, scission distance, selective channel scission model, effective fission barrier correction, nuclear deformation, fission cross-section

## Abstract

This study proposes a novel theoretical approach to understanding the statistical–mechanical similarities between nuclear fission phenomena and semiconductor physics. Using the Hill–Wheeler formula as a quantum mechanical distribution function and establishing its correspondence with the Fermi–Dirac distribution function, we analyzed nuclear fission processes for nine nuclides (^232^Th, ^233^U, ^235^U, ^238^U, ^237^Np, ^239^Pu, ^240^Pu, ^242^Pu, ^241^Am) using JENDL-5.0 data.

## 1. Introduction

Since its discovery by Hahn and Strassmann in 1938, nuclear fission has remained a central research topic in nuclear physics. While phenomenological understanding has advanced through liquid drop models and shell corrections, the statistical–mechanical foundation of nuclear fission remains incompletely understood.

In particular, while experimental data on fission product yield distributions has accumulated, theoretical prediction remains challenging. Even for major fission reactions in nuclear reactors, such as ^235^U and ^239^Pu, a complete theoretical description of mass and charge distributions remains elusive.

Meanwhile, in semiconductor physics, the description of electronic states based on Fermi–Dirac statistics has been highly successful. In particular, phenomena such as band gaps and carrier transport are precisely understood through statistical–mechanical approaches using the Fermi–Dirac distribution function.

This research proposes a novel theoretical approach based on the insight that there might exist statistical–mechanical similarities between nuclear fission phenomena and semiconductor physics, which appear to be completely different physical systems. Specifically, we reinterpret the Hill–Wheeler formula as a quantum mechanical extension of the Fermi–Dirac distribution function and attempt to provide a statistical–mechanical description of nuclear fission phenomena.

The characteristics of this approach are:Reinterpretation of the Hill–Wheeler formula as a quantum mechanical distribution function for nuclear fissionEstablishment of a systematic method to determine the Fermi energy for fission fragmentsPresentation of a statistical–mechanical interpretation of prompt neutron spectra

The structure of this paper is as follows. Section 2 presents an overview of the statistical–mechanical nuclear fission model, and Section 3 explains the specific calculation method using the Hill–Wheeler formula. Section 4 presents calculation results and discusses their physical significance, while Section 5 discusses the similarities between neutron spectra and semiconductor physics. Finally, Section 6 presents conclusions and future prospects.

This study is a substantially revised and academically reorganized version of the author’s previously published works [1,2,3]. In the earlier publications, $E_{x}$ was interpreted as the excitation energy; however, in this paper, $E_{x}$ is redefined as the Fermi energy, prompting a fundamental overhaul of both the theoretical framework and the numerical analysis approach. Specifically, new insights have been gained into the energy spectrum and cross-section analyses within the statistical model of nuclear fission. Another notable feature of this work is the expansion toward a more versatile numerical method, achieved by utilizing the optimization results obtained under Mathematica ver.11.2 and performing additional analyses and visualizations in the latest version (ver.14.1). In this paper. We also revisit the theoretical background that was not thoroughly addressed in the previous works [1,2,3], and discuss the physical implications of the analyses carried out using the newly redefined $E_{x}$. Through this approach, we aim to establish a new foundation for understanding fission phenomena from perspectives that were not addressed in earlier interpretations.

### Originality of This Research (Introduction of Statistical–Mechanical Interpretation)

Based on this background, we extend the statistical treatment of nuclear fission based on the Selective Channel Scission (SCS) model [4,5,6,7,8,9], which forms the foundation of this research, and present the following novel contributions.


**Extension from Mass Distribution to Charge Distribution:**
While the original theory (SCS model) primarily focused on calculating mass number distributions, this paper applies the model to charge-distribution calculations, achieving good agreement with experimental data. In particular, we demonstrate for the first time, to the author’s knowledge, that$$R_{\mathrm{eff}}\propto Z_{1}\times Z_{2}$$
meaning that the charge-distribution results show a proportionality to the product of atomic numbers of the two fission fragments. This conclusion supports the “intuitive” result that the fission distance is proportional to $Z_{1}Z_{2}$.
**Reinterpretation of $E_{x}$ as Fermi Energy:**
Previously, in the SCS model, $E_{x}$ was often treated as an “excitation energy”. In this research, by considering $E_{x}$ as a **Fermi energy**, we estimate different $E_{x}$ values for each fission channel, demonstrating more consistent reproduction of experimental distributions.
**Beyond the Hill–Wheeler Equation as Mere Transmission Formula:**
The Hill–Wheeler equation was originally known as a “barrier transmission probability”. This research explicitly shows its formal correspondence with the *Fermi distribution*, reinterpreting it as a “quantum mechanical extended Fermi distribution”. This new perspective clearly positions it as a distribution function for fission fragment production probability.
**Liberation from Large-Scale Computations:**
Previous SCS model studies required large-scale computations on supercomputers. This research presents a methodology using *Mathematica* that achieves comparable accuracy through relatively simple numerical analysis. This practical advantage makes the theory more accessible for introduction and verification.
**“Density of States × Fermi Distribution” Representation of Prompt Neutron Energy Distribution:**
Prompt neutron energy spectra were traditionally treated empirically using Maxwell or Watt distributions. This research focuses on describing them as a product of the density of states and Fermi–Dirac distribution, demonstrating that this can be theoretically explained using a quantum statistical model based on the Hill–Wheeler equation.

Through these points, this research generalizes the conventional SCS model, clarifies the *statistical* interpretation of the Hill–Wheeler formula, and organizes numerical methods in a form that is more reproducible for many researchers.

## 2. Overview of Statistical–Mechanical Nuclear Fission Model

While various approaches have been proposed for the theoretical description of nuclear fission phenomena, many require complex parameter adjustments. Among these, the Selective Channel Scission (SCS) model is noteworthy for its ability to predict fission yields with minimal parameters.

### 2.1. Basic Framework of the Selective Channel Scission Model

The SCS model, proposed by Takahashi, Ohta, et al. in 2001 [4,5], calculates yields by decomposing the nuclear fission process into individual channels and determining the fission barrier $E_{fi}$ for each channel. At the core of this model is the Hill–Wheeler formula [10,11,12,13]:(1)$$\begin{array}{c} f\left(E\right)=\frac{1}{1+exp[\frac{2\pi \Delta E}{\hbar \omega_{f}}]}, \end{array}$$
where the energy difference $\Delta E_{i}$ is defined as:(2)$$\begin{array}{c} \Delta E_{i}=E_{fi}-E_{x}, \end{array}$$
leading to the transmission probability:(3)$$\begin{array}{c} T_{i}\left(E_{x}\right)\approx \frac{1}{1+exp[-0.218\kappa \sqrt{\mu_{m}}\Delta E_{i}]}, \end{array}$$

### 2.2. Introduction of Statistical–Mechanical Interpretation

In this research, we propose a new statistical–mechanical interpretation of the Hill–Wheeler formula in the SCS model. Specifically, we reinterpret Equation (Equation 1) as a quantum statistical–mechanical distribution function for nuclear fission, introducing the following correspondences:$E_{x}$: Corresponds to Fermi energy (chemical potential)$E_{fi}$: Fission Barrier$\frac{\hbar \omega_{f}}{2\pi }$: Effective energy from quantum oscillations near the fission barrier (corresponding to kT)

Based on this interpretation, systematic analysis using JENDL-5.0 [14,15] nuclear fission yield data revealed the following important findings:The effective nuclear fission distance ($R_{eff}$) is proportional to the product of the charges of the fission fragments $\left(Z\right)\left(Z_{parent}-Z\right)$The effective fission distance shows maximum values in the symmetric fission region ($Z=Z_{parent}/2$) at the atomic nuclear scale (approximately 1 fm)The Fermi energy distribution has values unique to each fission fragment, with maximum values consistent with fission barrier energies.

## 3. Calculation Using the Hill–Wheeler Formula in Nuclear Fission Statistical Model

Here, we present the calculation conditions, results, and equations.

### 3.1. Calculation Conditions

The experimental data and target nuclei used in the calculations are as follows (Table 1):

It is noteworthy that this statistical model requires only the experimental data of fission product charge-distribution yields, neutron separation energy, incident neutron kinetic energy, and an average number of prompt neutrons. No adjustable parameters are needed.

### 3.2. Reasons for Considering the Hill–Wheeler Formula as Nuclear Fission’s Quantum Mechanical Distribution Function

The Hill–Wheeler equation has traditionally been introduced as a “formula for calculating the **transmission probability**” when approximating the nuclear fission barrier with a harmonic oscillator. However, in this research, we propose a framework to reinterpret the Hill–Wheeler equation, shown in the following equation, as a ***quantum mechanical occupation probability** with the same form as the Fermi–Dirac distribution*.(4)$$T\left(E\right)\;=\;\frac{1}{\;1+\exp\;\left[\frac{2\pi }{\hbar \;\omega_{f}}\;\left(E_{fi}-E_{x}\right)\right]}\;,$$
where $\omega_{f}$ is the angular frequency near the barrier, $E_{fi}$ is the nuclear fission barrier energy, and $E_{x}$ is the Fermi energy.

Below, we present four reasons for positioning the Hill–Wheeler equation not merely as a method for calculating barrier transmission probability but as a **quantum statistical distribution**.

#### 3.2.1. Origins in Quantum Many-Body Problem Treatment: Based on Generator Coordinate Method (GCM)

The Hill–Wheeler equation is derived from the superposition of *quantum mechanical many-body wave functions* in the **Generator Coordinate Method (GCM)**, which uses collective coordinates of the atomic nucleus (such as deformation parameters). In the process of calculating WKB approximation and tunnel effects, **interference terms and probability amplitude phases** contribute, resulting in an occupation probability that takes the form of a **sigmoid function** [10,16]:$$\frac{1}{\;1+\exp[\ldots ]}\;$$

For mere classical barrier transmission (e.g., Gamow factor), there would only be exponential decay of the form exp(−G) without becoming a sigmoid function. Therefore, it is natural to interpret the Hill–Wheeler Equation (Equation 4) as an occupation probability arising from the **combination of quantum mechanical amplitudes**.

#### 3.2.2. Calculation Results Converge to Values Reminiscent of “Fermi Levels”

In the Mathematica calculations discussed later in this research, energies corresponding to approximately 50–90% of the barrier energy $E_{f}$ appear as **critical points (boundaries between occupied and unoccupied states)** for each fission channel. Rather than viewing this as “partial excitation energy”, it is more systematically explicable when treated as a **Fermi level (chemical potential)**. Indeed, the behavior where *occupation probability changes significantly at certain energy boundaries* is typical of Fermi–Dirac-type statistical distributions and is consistent with the Hill–Wheeler sigmoid function.

#### 3.2.3. Correspondence with Prompt Neutron Spectra: Characteristics of Quantum Statistical Distribution

The prompt neutron spectrum from nuclear fission takes the form:(5)$$\chi \left(E\right)\;\propto \;\sqrt{E}\;\exp\left(-\alpha E\right)$$

This corresponds well with the product of the **density of states ($\sqrt{E}$)** and **quantum statistical occupation probability (exponential factor)**. The exp(−αE) term seen in Boltzmann and Fermi–Dirac distributions is difficult to derive from classical barrier transmission alone but can be naturally reproduced when considering that **neutrons (or protons) in the nucleus occupy states according to quantum statistics**. The Hill–Wheeler sigmoid function can be viewed as a type of **quantum occupation function** that supports this probability distribution. This point will be discussed in detail later using figures.

#### 3.2.4. Formal Equivalence with Fermi–Dirac Distribution: $\left(k_{B}T\right)↔\left(\hbar \nu \right)$

The Fermi–Dirac distribution is written as:(6)$$f_{\mathrm{FD}}\left(E\right)\;=\;\frac{1}{\;1+\exp\left[\frac{E-\mu }{k_{B}T}\right]}$$
where $k_{B}T$, the average energy scale in thermal equilibrium systems, appears in the denominator.

On the other hand, when defining the “frequency” in the harmonic oscillator approximation near the nuclear fission barrier in the Hill–Wheeler Equation (Equation 4) as:(7)$$\nu \;=\;\frac{\omega_{f}}{2\pi }$$
it can be written in a more intuitively understandable form:(8)$$T\left(E\right)=\frac{1}{1+\exp\left[\frac{E_{fi}-E_{x}}{\hbar \nu }\right]}$$
where ℏν appears as the **energy scale**. This leads to the simple and elegant correspondence:$$E_{x}\;↔\;\mu \;,\;\hbar \nu \;↔\;k_{B}T$$

Since the mathematical forms are **completely identical sigmoid functions**, there is no essential difference in reinterpreting the Hill–Wheeler equation as a **Fermi–Dirac type quantum mechanical occupation probability**.

##### Energy Scales as Common “Threshold” Parameters

ℏν: Effective energy increment brought about by **quantum oscillations** near the fission barrier$k_{B}T$: Energy dispersion brought about by **thermal motion** in many-particle systems

Both function as **threshold parameters** that determine “**at what energy difference the probability (distribution) changes rapidly**”. Indeed, both distribution formulas take the form of a **sigmoid function**:$$\frac{1}{1+\exp\;\left[\frac{\left(\mathrm{Energy}\;\mathrm{difference}\right)}{\mathrm{Scale}}\right]}$$
where the value transitions rapidly between 0 and 1 depending on whether the energy difference is sufficiently larger or smaller than this “scale”.

However, the **origins and interpretations** of ℏν and $k_{B}T$ are different:ℏν: Energy from quantum oscillations (harmonic oscillator model) of a **single nucleus or near-barrier region**.$k_{B}T$: Average (typical) energy in **thermal equilibrium**. Reflects the statistical–mechanical temperature of many-particle systems.

In this work, since atomic nuclei are not necessarily large thermal equilibrium systems, we take the position that the Hill–Wheeler formula using ℏν (quantum mechanical scale) in the denominator is **more suitable for describing single-nucleus fission probability**. However, since the **mathematical form** coincides with the Fermi distribution, we find it useful to show both in comparison.

Furthermore, since nuclear fission systems are not necessarily in thermal equilibrium, a description using $k_{B}T$ may not always be appropriate. On the other hand, the Hill–Wheeler formula using ℏν directly reflects the quantum mechanical properties of a single nucleus, providing a more suitable description. This interpretation is supported by the following calculation results:Correspondence between Fermi energy distribution and nuclear fission barrier energyAgreement of fission distance at the atomic nuclear scale (approximately 1.0–1.2 fm)Systematic reproduction of fission fragment charge distributions

Based on these four points, we can establish the view that “the Hill–Wheeler equation can be treated as a **type of quantum statistical distribution** while simultaneously being a transmission probability equation”. By interpreting the Hill–Wheeler equation, which has often been viewed merely as a barrier transmission equation in conventional models (including the SCS model), as a **“statistical distribution determining nucleon occupation/non-occupation”**, we can more naturally explain the behavior of nuclear fission cross-sections, including asymmetric nuclear fission, using concepts such as **Fermi levels and (proton) holes**, as discussed next. This represents our research’s **new perspective and contribution**.

**Note that the detailed derivation process of the Hill–Wheeler equation (detailed calculations from the one-dimensional Schrödinger equation through WKB approximation and Airy function connection to obtain the sigmoid function) is shown in the Appendix A.** Here, we will focus our discussion on the similarity with the Fermi-Dirac distribution, using only the fact that Equation (Equation 4) ultimately takes the form $T\left(E\right)=\frac{1}{1+\exp\left[\frac{2\pi (V_{0}-E)}{\hbar \omega_{f}}\right]}$.

##### (Supplementary Note) Points Difficult to Explain Under the Traditional “Barrier Transmission Coefficient” Interpretation

The Hill-Wheeler equation has long been treated as a **barrier transmission coefficient** (quantum mechanical tunneling probability). While this interpretation is certainly correct, in the following cases, the perspective of *mere transmission probability* alone is either insufficient for explanation or often becomes unnecessarily complicated.


**Correspondence with Prompt Neutron Spectra:**
The prompt neutron spectrum from nuclear fission is often summarized as $\chi \left(E\right)\propto \sqrt{E}\;\exp\left(-\alpha E\right)$. This form can be naturally interpreted as a product of **density of states** ($\sqrt{E}$) and **quantum statistical occupation probability** (exp(−αE)), but when emphasizing only the “barrier transmission coefficient”, it becomes less clear why this exponential term (exp) corresponds to “statistical occupation”. On the other hand, if we interpret the Hill–Wheeler equation as a *quantum statistical–mechanical distribution function* (see Figure 1 for illustration), it can be clearly explained as “occupation probability with ℏν as the energy scale”, in analogy to an energy band diagram isomorphic to spontaneous emission in semiconductor theory.
**Correspondence Between “Ex” Calculation Results and Chemical Potential (Fermi Level):**
In the numerical calculations of this study (see the Mathematica code discussed later), we obtained the notable result that for each fission channel, $E_{x}$ converges to 50–90% of the barrier energy $E_{fi}$. Under the conventional “barrier transmission probability interpretation”, $E_{x}$ can only be viewed as “excitation energy”, and it cannot adequately explain why it is around 50–90% of $E_{fi}$. However, through the interpretation as “quantum statistical probability” presented in this study, $E_{x}$ can be understood as energy equivalent to the **chemical potential (Fermi level)**, naturally explaining its relative relationship with the fission barrier.
**Complete Identity of Hill-Wheeler Equation and Fermi-Dirac Distribution:**
Given that the mathematical form $\frac{1}{1+e^{\left[\ldots \right]}}$ is *exactly identical* as a logistic function, it would be rather unnatural to interpret Hill-Wheeler only as a “barrier transmission coefficient” while denying its quantum statistical–mechanical properties. In fact, when viewing the Hill-Wheeler equation as a transmission probability formula, it is vaguely described within the category of “quantum mechanical tunneling probability”, but looking at its logistic form, it shows **precisely the same structure as “Fermi-Dirac type occupation probability”**, and interpreting it as a *quantum statistical distribution* is more concise and systematic.

As shown above, by extending beyond the conventional understanding of barrier transmission coefficient (quantum tunneling probability) (Of course, this is a legitimate interpretation of the Hill-Wheeler equation, but it remains only one aspect.) and positioning the Hill-Wheeler equation as a **quantum statistical distribution**, the following benefits emerge:Physical intuition obtained from the perspective of prompt neutron spectra and nucleon occupation probabilities becomes easier to organizeThe meaning of $E_{x}$ corresponding to the Fermi level can be naturally understoodAnalogy with band structure in semiconductor theory can be actively utilized
These perspectives are useful for deepening subsequent numerical analysis and interpretation of fission mechanisms from a **“statistical–mechanical viewpoint”**.

#### 3.2.5. Schematic Diagram of Asymmetric Nuclear Fission Through Analogy with Semiconductor Theory

Figure 2 is a schematic diagram that enables an intuitive understanding of particle motion and energy barriers during asymmetric nuclear fission by corresponding fission fragments to semiconductor PN junctions [3].

In semiconductor theory, photons are considered to be emitted through the diffusion and recombination of “electrons” and “holes”. In nuclear fission theory, this is replaced with the concept that neutrons are emitted through the diffusion and recombination of “protons” and “(proton) holes”. Furthermore, by considering that the semiconductor band gap corresponds to the nuclear fission barrier, the Hill–Wheeler equation functions as a mathematical formula equivalent to the Fermi distribution function. These correspondences are shown in Table 2.

Figure 2 (bottom) shows an example where La57 (high-charge fragment) corresponds to an N-type semiconductor and Br35 (low-charge fragment) corresponds to a P-type semiconductor. It illustrates the process of protons (electrons) and holes diffusing and drifting across the “depletion layer” in the barrier’s central region, where neutron (photon) emission occurs if recombination takes place. Meanwhile, Figure 2 (top) shows a typical PN junction model in semiconductor theory, contrasting how photons are emitted through electron and hole recombination.

As described above, just as the **Fermi–Dirac distribution** determines carrier distribution in PN junctions, the **Hill–Wheeler equation** governs the transmission probability (or production cross-section) of protons and holes in nuclear fission, demonstrating a common mathematical structure.

As will be shown in detail through comparison with charge-distribution experimental data, these correspondences provide insights for understanding the **mechanism of asymmetric nuclear fission** from the perspective of semiconductor theory.

Furthermore, by utilizing this correspondence relationship and performing calculations using charge-distribution experimental data, we can determine the **pseudo-Fermi energy as energy $E_{x}$ in Equation (Equation 4) for each fission fragment**, shown by the dashed lines in the figure. This leads to estimating the nuclear fission barrier energy corresponding to the “band gap”.

Through this analogy, we believe we can understand the mechanisms of asymmetric nuclear fission and the process of prompt neutron emission within a framework inspired by semiconductor theory.

This analogy is not merely “hand-waving” but is supported by **(a) mathematical form equivalence** and **(b) quantitative consistency with experimental data**, providing novelty that complements conventional fission models such as the SCS model.

### 3.3. Derivation of the Nuclear Fission Statistical Model

Here, we derive the equations for the nuclear fission statistical model from the Hill–Wheeler Formula (Equation 4), which we consider to be an advanced version of the distribution function.

Since Equation (Equation 4) was derived assuming a harmonic oscillator, we maintain this assumption. We consider the atomic nucleus as a harmonic oscillator where fission fragments ($FP_{1}$ and $FP_{2}$) are connected by an ideal spring, as shown in Figure 3. (Since we are dealing with charge-distribution experimental data, we use charges $Z_{1}$ and $Z_{2}$ rather than masses for the fission fragment harmonic oscillator).

Restating the Hill–Wheeler formula:(9)$$\begin{array}{c} f\left(E\right)=\frac{1}{1+exp[\frac{2\pi \Delta E}{\hbar \omega_{f}}]}, \end{array}$$

This equation was derived by Hill and Wheeler from the Schrödinger equation assuming a harmonic oscillator [10], where the harmonic oscillator potential is:(10)$$\begin{array}{c} V\left(r\right)=V_{0}-\frac{1}{2}\mu_{z}{\omega_{f}}^{2}r^{2}, \end{array}$$

The $\omega_{f}$ in the denominator is the angular frequency of the harmonic oscillator, and with $k_{f}$ as the force constant: (11)$$\begin{array}{c} \omega_{f}=2\pi \nu =2\pi \sqrt{\frac{k_{f}}{\mu_{z}}}, \end{array}$$

The reduced mass $\mu_{Z}$ uses charges $Z_{1},Z_{2}$ as shown in Figure 3:(12)$$\begin{array}{c} \mu_{Z}=\frac{Z_{1}*Z_{2}}{\left(Z_{1}+Z_{2}\right)}, \end{array}$$

Thus, Equation (Equation 9) becomes: (13)$$\begin{array}{c} f\left(E\right)=\frac{1}{1+exp\left[\frac{2\pi \Delta E}{\hbar \sqrt{\frac{k_{f}}{\mu_{z}}}}\right]}, \end{array}$$

Since $\sqrt{\mu_{Z}}$ is the product of charges $Z_{1},Z_{2}$ according to Equation (Equation 12) and describes a parabola, setting its maximum value as $\sqrt{\mu_{Zmax}}$: (14)$$\begin{array}{c} \sqrt{\mu_{Z}}=\sqrt{\mu_{Zmax}}*\sqrt{\frac{\mu_{Z}}{\mu_{Zmax}}}, \end{array}$$

Then: (15)$$\begin{array}{c} f\left(E\right)=\frac{1}{1+exp\left[\frac{2\pi \sqrt{\frac{\mu_{Z}}{\mu_{Zmax}}}\Delta E}{\hbar \sqrt{\frac{k_{f}}{\mu_{zmax}}}}\right]}, \end{array}$$

Let us compare this with the Fermi–Dirac distribution function (restating Equation (Equation 6)):(16)$$\begin{array}{c} f\left(E\right)=\frac{1}{1+exp[\frac{\Delta E}{kt}]}, \end{array}$$

The value corresponding to kt in the Fermi–Dirac distribution function (Equation 16) is $\hbar \sqrt{\frac{k_{f}}{\mu_{zmax}}}$ in Equation (Equation 15). Considering what corresponds to this temperature energy kt, we propose it is the incident neutron energy. The reason for this is that just as the distribution state in a container (solid, semiconductor, etc.) changes with temperature energy kt in the Fermi–Dirac distribution function (Equation 16), similarly, the distribution state in the container (atomic nucleus) changes with incident neutron energy in distribution function (Equation 15).

Therefore, with Sn = neutron separation energy and Kn = neutron kinetic energy: (17)$$\begin{array}{c} \hbar \sqrt{\frac{k_{f}}{\mu_{zmax}}}=Sn+Kn, \end{array}$$

This leads to: (18)$$\begin{array}{c} f\left(E\right)=\frac{1}{1+exp\left[\frac{2\pi \sqrt{\frac{\mu_{Z}}{\mu_{Zmax}}}\Delta E}{Sn+kn}\right]}, \end{array}$$

Here, ΔE is considered to be the difference between $E_{fi}$ (fission barrier energies of fission fragments) and $E_{x}$ (Fermi energy):(19)$$\begin{array}{c} \Delta E_{i}=E_{fi}-E_{x}, \end{array}$$

In our nuclear fission calculations, both $E_{x}$ (which becomes Fermi energy) and $E_{fi}$ vary with the elements constituting nuclear matter (we will later change $E_{x}$ to $E_{xi}$). The calculation of $E_{fi}$ will be explained in the next section, and the variation of $E_{x}$ will be discussed in later considerations.

Thus, the final statistical model equation using the Hill–Wheeler formula becomes: (20)$$\begin{array}{c} f_{i}\left(E\right)=\frac{1}{1+exp\left[\frac{2\pi \sqrt{\frac{\mu_{Z}}{\mu_{Zmax}}}\Delta E_{i}}{Sn+kn}\right]}, \end{array}$$

### 3.4. Calculation Process from Nuclear Fission Statistical Model to Charge Distribution

Here, we show the calculation process for obtaining charge distribution using Equation (Equation 20) derived from the Hill–Wheeler formula.

Near the potential barrier peak shown in Figure 4, various numerical calculations have been attempted, but no definitive method exists. Currently, it is thought that shell effects may form multiple barriers, but in our case, the nuclear Fermi energy obtained for each fission fragment will have different values, ultimately showing a structure that forms fission isomers. However, for calculation purposes, we assume a single-barrier structure.

The validity of this assumption is demonstrated by later calculation results showing that the fission energy used in calculating charge distribution is predominantly due to the Coulomb force between two fission fragments, with nuclear-force effects being comparatively small, making the single-barrier assumption acceptable.

To reiterate, while the potential strictly forms multiple barriers, since the dominant contribution comes from the Coulomb force between two fission fragments, treating it as a single barrier with small perturbations is justified.

As shown in Figure 4, when two fission fragments are separated sufficiently far apart, the potential equals the Coulomb potential. Therefore, we assume that fission occurs at the internuclear distance where the bare fragment Coulomb potential corresponds to the channel-dependent fission barrier. We call this internuclear distance the effective fission distance $R_{eff}$, and define it using fission parameters k1 and k2 as follows:(21)$$\begin{array}{c} R_{eff}≒k1*\left(Z1*Z2+k2\right), \end{array}$$
where Z1 and Z2 are the charges of the fission fragments after fission.

The reason for introducing these fission parameters k1 and k2 is that through the deformation process leading to fission, the nuclear shape at the time of fission is expected to elongate in proportion to the charges Z1 and Z2 of the spherical fission fragments, as shown in Figure 3. Therefore, the distance between fission fragments, accounting for this elongation, should be proportional to the product of fragment charges Z1∗Z2 with parameters k1 and k2 considered. These parameters k1 and k2 can be automatically calculated from experimental data and do not need to be determined manually.

The channel-dependent fission barrier $E_{fi}$ is determined by subtracting the Q-value for each fission reaction from the value of the interfragmentary Coulomb potential (effective Coulomb energy) at the effective fission distance $R_{eff}$:(22)$$\begin{array}{c} E_{fi}=E_{c}^{\prime }-Q, \end{array}$$

The first term on the right side, the Coulomb potential $E_{c}^{\prime }$, is:(23)$$\begin{array}{c} E_{c}^{\prime }=1.44*Z1*Z2/R_{eff}, \end{array}$$

These calculation Formulas (Equation 22) and (Equation 23) are the same as in the SCS model [4,5,6,7,8,9].

Finally, the charge-distribution yields of each fission product are calculated by substituting these values into the nuclear fission statistical model Equation (Equation 20) for each fission reaction and then computing:(24)$$\begin{array}{c} T_{i}\left(E_{xi}\right)=\frac{\sum_{j}f_{i}\left(E_{xi}^{j}\right)}{\sum_{j}f_{i}\left(E_{xi}^{j}\right)+\sum_{j}f_{i}\left(E_{xi+1}^{j}\right)+......}, \end{array}$$

Here, $\sum_{j}f_{i}\left(E_{xi}^{j}\right)$ means summing over all possible isotopes (denoted by subscript *j*) of the fission fragments that can be produced in nuclear fission. For example, in the case of the atomic number *i* = 56 (Ba56), this would mean summing all possible values for fission products from Ba56  114 to Ba56  153.

### 3.5. Specific Calculations

In this study, we used charge-distribution experimental data as calculation data, i.e., calculation conditions, and performed the following three calculations using the nuclear fission statistical model:**Calculation of Fission Distance Between Two Fission Fragments**In Equation (Equation 19), $\Delta E_{i}=E_{fi}-E_{x}$, we set the Fermi energy $E_{x}=0$ and solved for the fission distance $R_{eff}$ in Equation (Equation 21) as an unknown variable within Equation (Equation 20). As shown in Figure 5, the results showed that the fission distance is proportional to the charges *Z* of the two fission fragments. Specifically, it is proportional to *Z* * (atomic number-*Z*) = *Z* * atomic number −$Z^{2}$.Furthermore, these results showed that this distance distributes within the range of approximately 1.0–1.2 fm. This agrees with values predicted by other theories.**Calculation of Charge Distribution for Zero Fermi Energy (Ground State)**Using the fission distance $R_{eff}$ obtained in step 1 above, we calculated theoretical values for charge distribution using Equation (Equation 20) with Fermi energy $E_{x}=0$ (ground state) in Equation (Equation 19), $\Delta E_{i}=E_{fi}-E_{x}$. Figure 6 shows a comparison between theoretical values and experimental data. As can be seen from the figure, even with Fermi energy $E_{x}=0$, theoretical values closely match experimental data.**Calculation of Fermi Energy for Each Fission Fragment**First, we modified Equation (Equation 19), $\Delta E_{i}=E_{fi}-E_{x}$ to $\Delta E_{i}=E_{fi}-E_{xi}$, assuming the existence of different Fermi energies $E_{xi}$ for different channels. Then, using the fission distance $R_{eff}$ obtained above, we solved for this $E_{xi}$ as an unknown variable by solving simultaneous equations for all possible patterns of compound nucleus splitting into two fission fragments to obtain the nuclear fission yield charge distribution. As shown in Figure 7, this resulted in obtaining Fermi energies $E_{xi}$ with different values for each element (fission fragment) constituting the compound nucleus.For detailed calculations, please refer to the attached Mathematica code.

## 4. Results and Discussion

In this section, we discuss the physical meaning and theoretical interpretation of the calculation results presented in the previous section.

### 4.1. Physical Significance of Effective Fission Distance Between Fragments

As a representative example, Figure 5 shows the calculation results for U235.

For U235, the effective fission distance $R_{eff}$ is expressed as:$$\begin{array}{c} R_{eff}=-0.517291+0.000824625\left(92.0016-Z\right)Z\;\left[\mathrm{fm}\right] \end{array}$$

Similar analyses were conducted for the other eight nuclides, all showing similar quadratic dependence: Th90  232 effective fission distance (500 keV) Reff=−0.457368+0.000812599(89.9987−Z)ZU92  233 effective fission distance (500 keV) Reff=−0.362683+0.000727996(92.9985−Z)ZU92  235 effective fission distance (0.0253 eV) Reff=−0.517291+0.000824625(92.0016−Z)ZU92  238 effective fission distance (500 keV) Reff=−0.339285+0.000731257(92.0027−Z)ZNp93  237 effective fission distance (0.0253 eV) Reff=−0.364831+0.000729022(92.995−Z)ZPu94  239 effective fission distance (0.0253 eV) Reff=−0.405216+0.000732862(94.0015−Z)ZPu94  240 effective fission distance (0.0253 eV) Reff=−0.396877+0.000729146(93.9991−Z)ZPu94  242 effective fission distance (0.0253 eV) Reff=−0.410242+0.000734583(94.004−Z)ZAm95  241 effective fission distance (0.0253 eV) Reff=−0.466525+0.000744743(95.0044−Z)Z

From these results, the following important characteristics have emerged:

1. For all nuclides, the effective fission distance is proportional to the product of fragment charges 2. The proportionality coefficient is nearly constant across nuclides (range: 0.000727–0.000825) 3. Maximum values appear in regions where each nuclide’s charges are nearly symmetric 4. The obtained effective fission distances are on the scale of atomic nuclear size (approximately 1.0–1.2 fm).

This charge dependence is thought to reflect the collective motion of nucleons during the fission process. In particular, the convergence of the proportionality coefficient to a narrow range suggests the existence of a universal characteristic in the fission mechanism.

These calculation results show that when the product of fragment charges $Z_{1}$ and $Z_{2}$ (atomic number-$Z_{1}$) is maximum, the Coulomb repulsion force reaches its maximum value, and the nuclear fission distance becomes maximum. This value was derived as a result of calculating the fission distance $R_{eff}$ as an unknown variable, but it agrees with the intuitive image of nuclear fission.

It is difficult to imagine that the distance at nuclear fission would be independent of charges $Z_{1}$ and $Z_{2}$. Although the exact process of nuclear fission cannot currently be confirmed experimentally, it is most natural and consistent with intuition to consider that it is proportional to the product of fragment charges $Z_{1}$ and $Z_{2}$, as shown in Figure 5.

Moreover, the calculation results show that the fission distance $R_{eff}$ is approximately 1.0–1.2 fm. While nuclear-force potential values cannot be directly determined experimentally, calculations using Woods-Saxon potential models and QCD lattice theory also indicate that nuclear forces weaken rapidly beyond a radius of about 1 fm. (As shown in Figure 8) [17].

This result does not contradict the finding that the fission distance $R_{eff}$ is approximately 1.0–1.2 fm. It suggests that nuclear fission occurs when nuclear forces weaken around 1.2 fm, and Coulomb repulsion between fragments becomes dominant.

As described above, for all nine analyzed nuclides, we found that the fission distance is proportional to the product of fragment charges Z1 and Z2, and $R_{eff}$ is approximately 1.0–1.2 fm. It is unlikely that such simple results would emerge by chance. Therefore, we consider this to be one piece of evidence supporting the validity of our nuclear fission statistical model (Equation 20), which assumes the Hill–Wheeler Formula (Equation 9) as the distribution function for nuclear fission.

### 4.2. Significance of Charge-Distribution Reproduction in Ground State

Figure 6 shows a comparison between theoretical predictions at zero Fermi energy (ground state) and experimental data for U235 as a representative example.

In this study, we revealed the state of nuclear fission Fermi energy through the following analysis process:

First, using nuclear fission yield charge-distribution experimental data from JENDL-5, we calculated the effective distance between fission fragments ($R_{eff}$). This revealed that this distance is proportional to the product of the charges of the two fission fragments. This proportional relationship was confirmed to hold for all analyzed nuclides.

Next, using the obtained effective distance, we derived theoretical curves using the Hill–Wheeler formula and performed fitting with experimental data. By taking the difference between the fitted values and those predicted by the original theoretical equation, we quantitatively determined the unique energy states possessed by each fission fragment.

In this analysis process, the correction values obtained through fitting represent the Fermi energies of each fission fragment.

Specifically, we calculated the Fermi energy $E_{x}$ of each fission fragment from the difference ΔE between the fission fragment’s energy level $E_{fi}$ and the correction value ($E_{x}=E_{fi}-\Delta E$).

A particularly noteworthy point is that, as shown in Figure 6, the charge distribution observed experimentally can be reproduced almost perfectly even when this Fermi energy is set to zero. This discovery provides the important insight that nuclear fission is primarily driven by the nucleus’s internal energy (Coulomb force between the two fragments), and the Fermi energy required for fission is surprisingly small compared to this self-energy.

### 4.3. Inhomogeneity of Fermi Energy Distribution

Figure 7 shows the relationship between Fermi energy distribution and charge distribution for U235 as a representative example.

From these calculation results, the following important characteristics of Fermi energy during nuclear fission have been revealed:

1. Fermi energy is not uniform across the entire nucleus but strongly depends on the fission fragment’s charge number 2. Its distribution shows a close correlation with the charge distribution 3. The maximum values are comparable to fission barrier energies.

A particularly noteworthy point is that these results show remarkable similarities with semiconductor physics. In semiconductors, it is known that the Fermi energy is approximately half the value of the band gap energy (for example, for Si with a band gap of 1.1 eV, the Fermi energy is about 0.5 eV). Similarly, in our calculations, the nuclear fission Fermi energy was found to be about 0.5–0.9 times the predicted fission barrier energy.

### 4.4. Fermi Energy and Fission Barrier

Table 3 shows a comparison between the maximum values of Fermi energy obtained from our nuclear fission statistical model and the generally predicted fission barrier energies.

This comparison has revealed important physical insights.

For U235, the average value of the calculated Fermi energy is 3.85 MeV, which is approximately 0.65 times the predicted fission barrier energy of 5.8–6.0 MeV. Similar ratios are observed for other nuclides, with calculated Fermi energies typically ranging from 0.5 to 0.9 times the predicted barrier energies.

For Pu239 and Pu240, the average values of calculated Fermi energies (4.36 MeV and 4.90 MeV, respectively) are about 0.74–0.82 times the predicted fission barrier energies (5.9–6.0 MeV), showing ratios similar to those observed in the relationship between band gap and Fermi energy in semiconductors.

However, for Pu242, the calculated average Fermi energy (6.38 MeV) exceeds the predicted fission barrier energy (about 6.0–6.2 MeV). This exceptional behavior might be attributed to the unique nuclear structure of Pu242 and requires further investigation.

The variation of Fermi energy with incident neutron energy also provides important insights. The calculation results for three energy regions — 0.0253 eV, 500 keV, and 14 MeV — show systematic changes for many nuclides. For example, in U235, the Fermi energy changes from 4.13 MeV at 0.0253 eV to 3.57 MeV at 500 keV and 3.85 MeV at 14 MeV. This can be interpreted as a phenomenon similar to the temperature dependence in semiconductors.

The significantly lower calculated values for U233 and Np237 (average values of 2.94 MeV and 2.96 MeV, respectively) compared to predicted values suggest strong influences of shell effects and pairing correlations in these nuclides. This can be considered analogous to the effect of impurity levels in semiconductors.

## 5. Neutron Spectrum ∝ Density of States × Fermi–Dirac Distribution Function

Finally, we demonstrate that the energy spectrum of prompt neutrons can be expressed as the product of the density of states and the Fermi–Dirac distribution function, similar to spontaneous emission of light in semiconductors.

### 5.1. Spectrum of Spontaneous Emission Light in Semiconductors

The spectral shape of spontaneous emission light in semiconductors can be expressed as shown in Figure 9 [18].

This spectral shape can be expressed mathematically as:(25)$$\begin{array}{cc} I(\hbar \omega ) & \propto \omega^{2}{\left(\hbar \omega \right)}^{1/2}exp\left(-\frac{\hbar \omega -E_{g}}{k_{B}T}\right)\\ \; & =\omega^{2}E^{1/2}exp\left(-\frac{\hbar \omega -E_{g}}{k_{B}T}\right) \end{array}$$

The low-energy side follows the energy distribution of the electron density of states, while the high-energy side reflects the Boltzmann distribution. In other words, in semiconductors, the energies of electrons and holes are distributed according to the Fermi–Dirac distribution function.

### 5.2. Energy Spectrum of Prompt Neutrons

The energy spectrum of prompt neutrons is shown in Figure 10.

As shown in the figure, this spectrum can be expressed as:(26)$$\begin{array}{cc} \chi (E) & =0.770E^{1/2}e^{-0.776E}\\ \; & \propto \mathrm{Density}\;\mathrm{of}\;\mathrm{States}\times \;\mathrm{Boltzmann}\;\mathrm{Distribution}. \end{array}$$

This equation has fundamentally the same form (although with different coefficients) as Equation (Equation 25) for semiconductor spontaneous emission light [19,20].

The energy distribution of fission products revealed in this research shows the following characteristics:In the low-energy region, it shows $\sqrt{E}$ dependence following the neutron density of statesIn the high-energy region, it shows exponential decay characteristic of the Boltzmann distributionAcross the entire energy range, it shows a distribution based on Fermi–Dirac statistics.

This behavior shows remarkable similarities with electron–hole distributions in semiconductor physics. In particular, the fact that the energy distribution of nucleons (protons and neutrons) follows Fermi–Dirac statistics is consistent with the picture shown in Figure 9, where the distribution can be expressed as the product of the density of states and the Fermi–Dirac distribution function. This similarity supports the validity of the statistical–mechanical approach proposed in this research.

## 6. Conclusions and Future Prospects

In this research, we conducted a theoretical investigation of statistical–mechanical similarities between nuclear fission phenomena and semiconductor physics. Particularly focusing on the formal similarity between the Hill–Wheeler formula and the Fermi–Dirac distribution function, we performed detailed analyses and obtained the following important findings:The effective distance between fission fragments is proportional to the product of their charges and distributes within the range of approximately 1.0–1.2 fm. This result is consistent with conventional theoretical predictions.By treating nuclear Fermi energy as an unknown variable and solving simultaneous equations for all fission channels, we were able to determine the Fermi energy distribution of each fission fragment under various energy conditions.The maximum values of these Fermi energies were found to be comparable to the conventionally predicted fission barrier energies. The relationship between these values shows similarities to the relationship between band gap and Fermi energy in semiconductor physics.We demonstrated that the energy spectrum of prompt neutrons can be expressed as the product of the density of states and the Fermi–Dirac distribution function, similar to spontaneous emission light in semiconductors.

These results reveal surprising statistical–mechanical similarities between nuclear fission phenomena and semiconductor physics, which appear to be completely different physical systems. These findings suggest the existence of common fundamental principles in the physical background of both systems, providing new perspectives for understanding nuclear fission phenomena.

Furthermore, in this research, we presented a theoretical framework that approximates the region near the nuclear fission barrier with a harmonic oscillator model and interprets the Hill–Wheeler equation as a quantum mechanical occupation probability with the same form as the Fermi–Dirac distribution. This approach not only naturally explains the charge distribution of fission fragments and prompt neutron spectra but also demonstrates that asymmetric nuclear fission phenomena can be intuitively understood through analogy with band structure in semiconductor physics. In this section, we discuss future challenges and prospects based on the results obtained thus far.

### 6.1. Precise Determination of Neutron Emission Numbers for Each Fission Fragment

In our current calculations, we have conducted theoretical analyses using average values for the number of neutrons emitted during nuclear fission reactions. However, to construct a more accurate theoretical model, we need to incorporate the actual neutron emission numbers that differ for each reaction pathway. For example, in(27)U235+n→FP1+FP2+xn+Q
where neutron incidence on U235 produces fission fragments (${\mathrm{FP}}_{1}$, ${\mathrm{FP}}_{2}$) and emitted neutrons (*x*), the number is likely to vary for each fission channel. Detailed evaluation of these variations would lead to a precise re-evaluation of nuclear fission barrier energies, consequently improving the prediction accuracy of charge distributions and energy spectra.

### 6.2. Multidimensional Analysis of Fermi Energy Distribution

Our current analysis primarily focuses on comparing maximum Fermi energy values with neutron separation energies. However, for a more precise prediction of nuclear fission frequencies, we need to consider that Fermi energy exists as a **continuous distribution** rather than a single maximum value and evaluate it **two-dimensionally** with neutron separation energy. Furthermore, by incorporating quantum effects such as energy level multiplicity, we expect to address excited states of fission fragments and detailed energy spectra of prompt neutrons. The introduction of these multidimensional analytical methods will likely deepen our statistical–mechanical understanding of nuclear fission phenomena.

### 6.3. Extension of Theoretical Model

For future development, it is important to verify how the “similarity with semiconductor physics” and “quantum statistical distribution interpretation of the Hill–Wheeler equation” revealed in this research can be consistent with other theoretical models. For instance, by incorporating our findings into Hauser-Feshbach statistical decay theory [21] and other multi-channel coupling models and applying them to a wider range of nuclides and incident neutron energy regions, we may be able to achieve a unified understanding of nuclear fission reactions as a whole. In this process, the extent to which the conceptual framework of semiconductor physics can be rigorously applied will also become a subject of theoretical and experimental verification.

As described above, while the results of this research bring new perspectives to the statistical–mechanical interpretation of asymmetric nuclear fission, they also provide a foundation for extending nuclear fission theory in a more microscopic and quantitative manner. Further elucidation of nuclear fission reaction mechanisms is expected through the refinement of neutron emission numbers for each nuclide and reaction channel, multidimensional analysis of Fermi energy distributions, and integration with existing models such as Hauser-Feshbach theory.

## Figures and Tables

**Figure 1 entropy-27-00227-f001:**
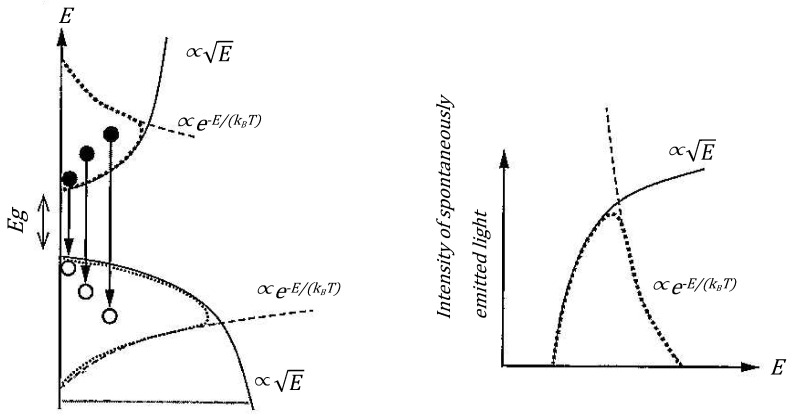
Conceptual diagram and emission spectrum of spontaneous emission light in semiconductors (excerpted from “Introduction to Optical Devices” by Professor Suemasu).

**Figure 2 entropy-27-00227-f002:**
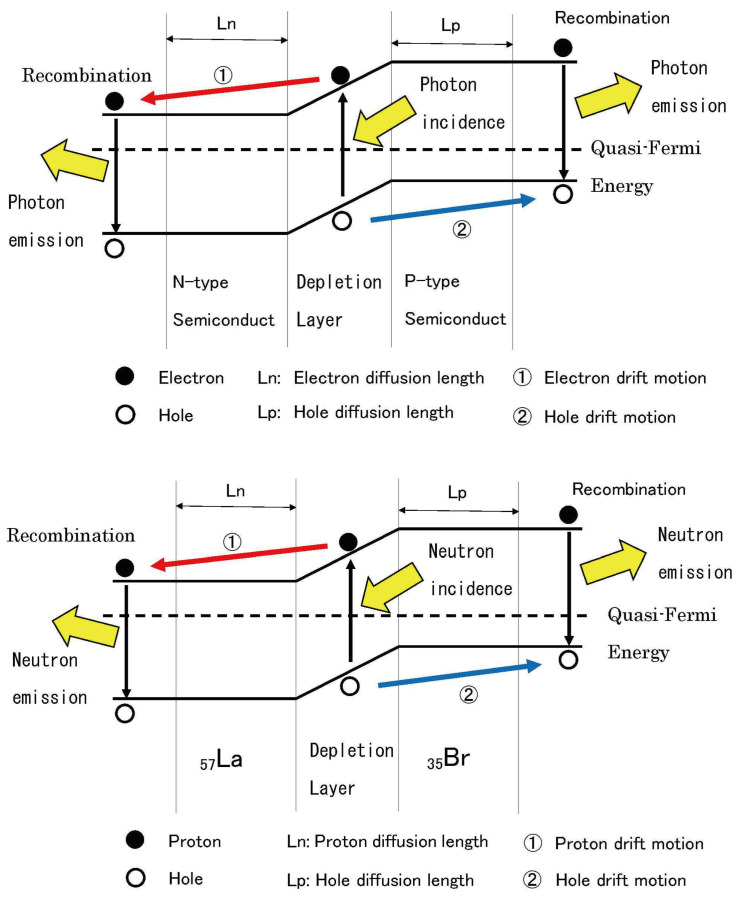
(**Top**) Schematic band diagram of PN junction in semiconductor theory; (**Bottom**) Schematic energy band diagram of asymmetric nuclear fission in nuclear theory.

**Figure 3 entropy-27-00227-f003:**
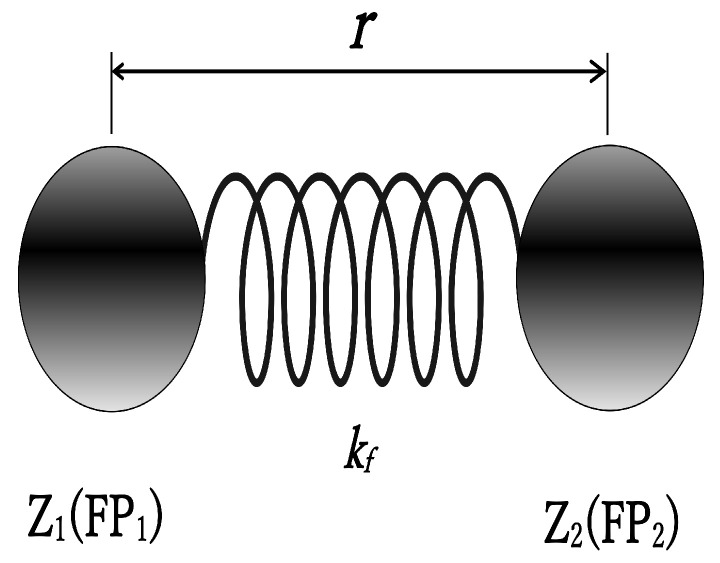
The atomic nucleus is modeled as a harmonic oscillator with fission fragments ($FP_{1}$ and $FP_{2}$) connected by an ideal spring.

**Figure 4 entropy-27-00227-f004:**
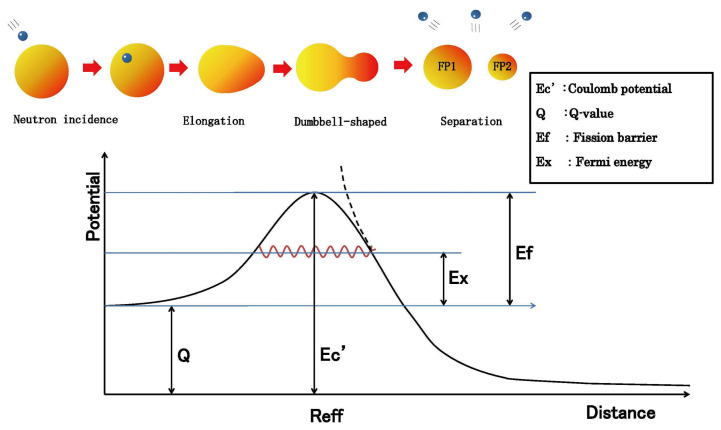
Relationship between nuclear fission process, potential energy, and effective fission distance $R_{eff}$.

**Figure 5 entropy-27-00227-f005:**
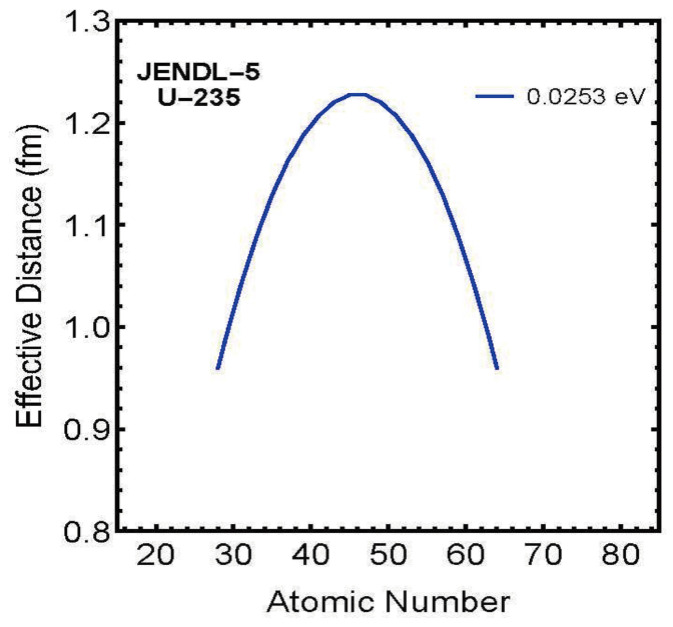
Charge number dependence of effective fission distance for U235 (incident neutron energy: 0.0253 eV).

**Figure 6 entropy-27-00227-f006:**
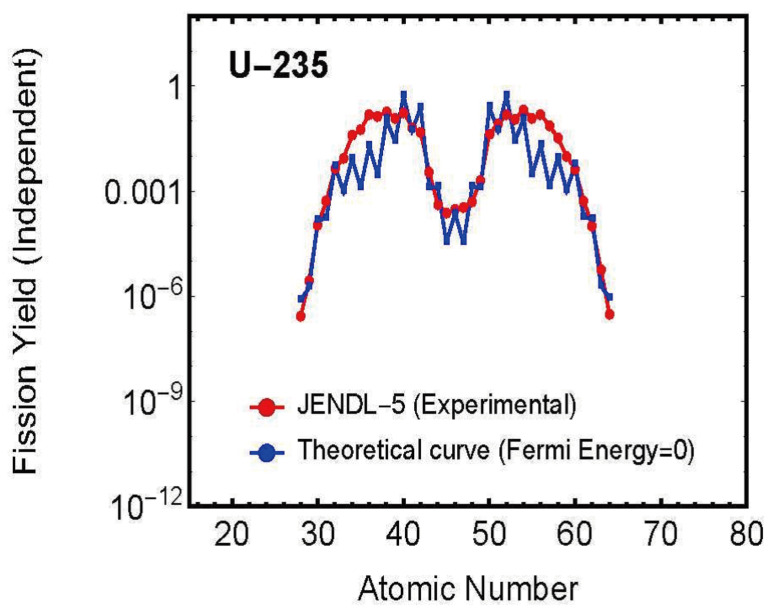
Charge distribution (theoretical curve) at zero Fermi energy (ground state) and experimental values for U235.

**Figure 7 entropy-27-00227-f007:**
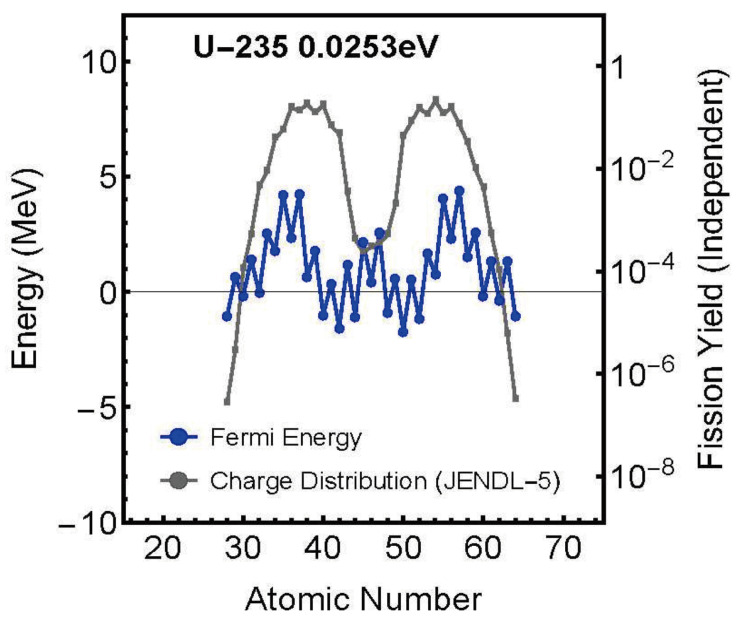
Relationship between Fermi energy distribution (blue line) and charge distribution (gray line) versus fission fragment charge number for U235 (incident neutron energy: 0.0253 eV).

**Figure 8 entropy-27-00227-f008:**
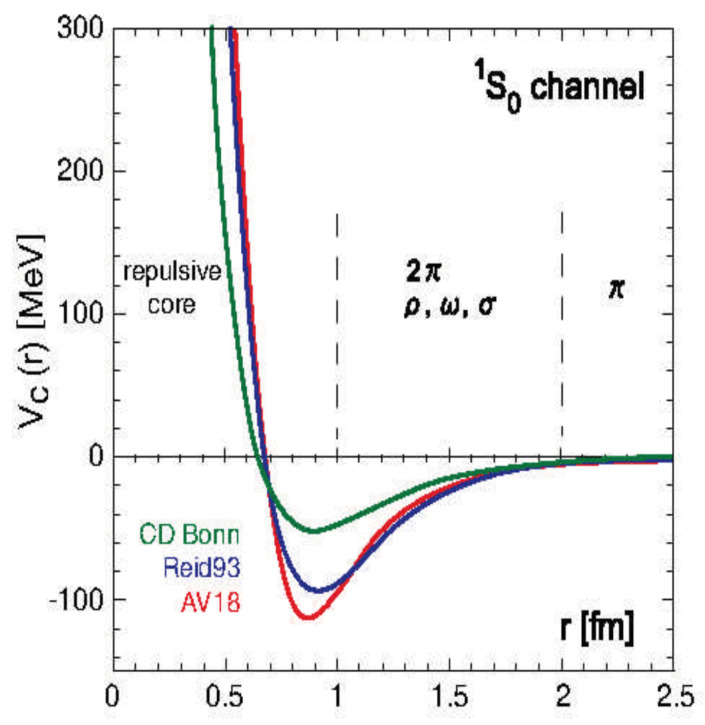
Relationship between atomic nuclear radius and potential energy.

**Figure 9 entropy-27-00227-f009:**
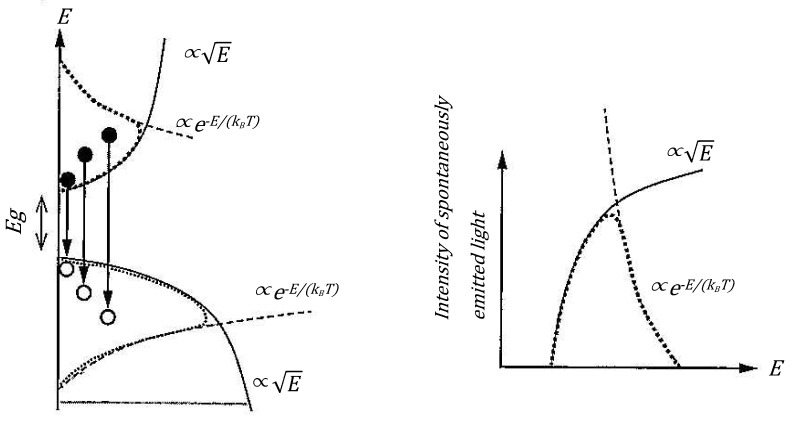
Conceptual diagram and emission spectrum of spontaneous emission light in semiconductors (excerpted from “Introduction to Optical Devices” by Professor Suemasu).

**Figure 10 entropy-27-00227-f010:**
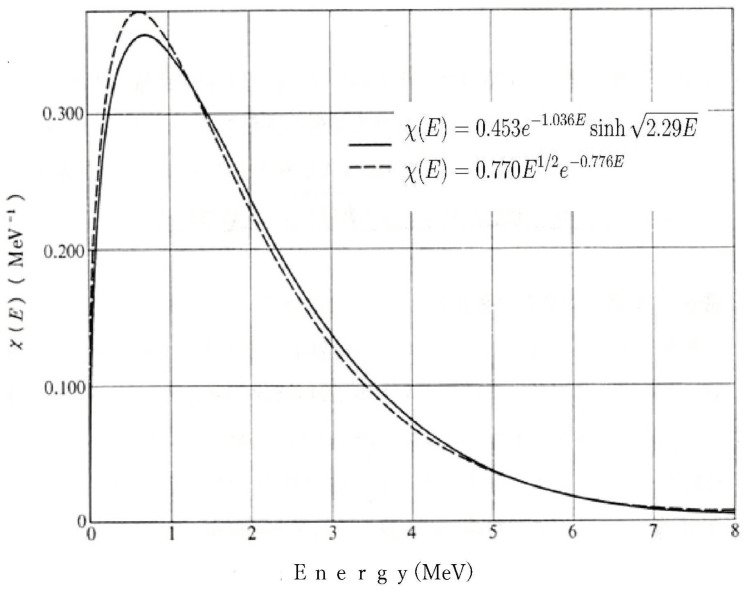
Energy spectrum of prompt neutrons (excerpted from Professor Abe’s website).

**Table 1 entropy-27-00227-t001:** Nine target nuclei and calculation conditions used in this study.

Target	Neutron	Incident Neutron	Average Number of Prompt
**Nucleus**	**Separation Energy**	**Energy**	**Neutrons (0.0253 eV/500 keV/14 MeV)**
Th232+n	4.7863 MeV	500 keV, 14 MeV	-/2.198/4.402
U233+n	6.8455 MeV	0.0253 eV, 500 keV, 14 MeV	2.497/2.933/4.521
U235+n	6.5455 MeV	0.0253 eV, 500 keV, 14 MeV	2.437/2.879/4.378
U238+n	4.8063 MeV	500 keV, 14 MeV	-/2.579/4.458
Np237+n	5.4882 MeV	0.0253 eV, 500 keV, 14 MeV	2.683/2.788/4.401
Pu239+n	6.5342 MeV	0.0253 eV, 500 keV, 14 MeV	2.875/3.242/4.891
Pu240+n	5.2415 MeV	0.0253 eV, 500 keV, 14 MeV	2.860/3.236/4.893
Pu242+n	5.0336 MeV	0.0253 eV, 500 keV, 14 MeV	2.936/3.276/4.921
Am241+n	5.5287 MeV	0.0253 eV, 500 keV, 14 MeV	3.209/3.453/4.972

**Table 2 entropy-27-00227-t002:** Correspondence between Semiconductor Theory and Nuclear Fission Theory.

Semiconductor Theory	Nuclear Fission Theory
Electron	Proton
(Electron) Hole	(Proton) Hole
Photon	Neutron
N-type Semiconductor	High-charge Fission Fragment La57 etc.
P-type Semiconductor	Low-charge Fission Fragment Ba35 etc.
Band Gap	Fission Barrier
Fermi Distribution Function	Hill–Wheeler Equation

**Table 3 entropy-27-00227-t003:** Comparison of maximum Fermi energy values and fission barrier energies.

Nuclide	Maximum Fermi Energy (MeV)				Generally Predicted
	**0.0253 eV**	**500 keV**	**14 MeV**	**Average**	**Fission Barrier (MeV)**
^232^Th	-	6.19	5.50	5.84	6.0∼6.3
^233^U	3.26	3.26	2.29	2.94	5.7∼6.0
^235^U	4.13	3.57	3.85	3.85	5.8∼6.0
^238^U	-	5.54	5.45	5.50	6.0∼6.2
^237^Np	3.43	3.16	2.29	2.96	5.8∼6.0
^239^Pu	4.72	4.25	4.12	4.36	5.9∼6.0
^240^Pu	5.36	4.84	4.51	4.90	5.8∼6.1
^242^Pu	6.48	5.98	6.69	6.38	6.0∼6.2
^241^Am	4.72	4.25	4.12	4.36	5.9∼6.1

## Data Availability

All nuclear fission yield data supporting the reported results were obtained from JENDL-5.0 [14,15]. No new data were created.

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
