# Peer review of "Application of the Hill-Wheeler Formula in Statistical Models of Nuclear Fission: A Statistical–Mechanical Approach Based on Similarities with Semiconductor Physics"

_entropy, 2025, doi:10.3390/e27030227_

Round 1
Reviewer 1 Report
Comments and Suggestions for Authors
Dear Editor,
I recommend that this paper will not be accepted for publication unless the author clearly explains what the novel contribution of his paper is.
The author discusses the Selective Model Scission (SMS) Model introduced in 2001 by Takahashi et al. to describe statistical results of fission processes. But this discussion is done using material already published by other authors, without any relevant novel contribution or insight of his own.
In the Introduction to his paper, the author tries to make a conceptual connection between the Hill-Wheeler distribution on which the SMS model is based and the Fermi-Dirac statistical distribution of fermions in thermal equilibrium. Still, other than some hand-waving, this identification is not sustained in any way, since the model's success in reproducing the collected experimental data does not rely at all on such identification.
Comments on the Quality of English LanguageThe paper is readable, but the English must be improved.
Reviewer 2 Report
Comments and Suggestions for Authors
Nice exercise of statistical physics and mathematical physics. Even if the method of analogy must be mastered with great attention and the author must be careful since the concepts and meanings in one model for a certain phenomenology are delicate to migrate in another phenomenology. Anyway the author did this task correctly and I fully agree with his results and conclusions.
Reviewer 3 Report
Comments and Suggestions for Authors
This research looks amateurish uns unprofessional.
It was partly written by IA, as the author condesses.
It is centres on the Hill-Whheler approach to statitical mechanocs, a beautogulr theory that is not described here.
The authors reduce it to Eqs. 1-3, that are not well defined.
Neither is the fission process.
There might be a hint of originality, but it needs more development.
I suggest REJECTION.
Round 2
Reviewer 1 Report
Comments and Suggestions for Authors
Dear Editor,
I cannot recommend that this paper will be accepted for publication.
The author insists that a major contribution of his paper is the insight gained from the identification of the Hill-Wheeler with the Fermi-Dirac distribution function and dedicates great effort to justify or explain it. However, besides the conceptual difficulties associated with such identification, neither I can find anywhere the 'workings' of such identification. All this discussion is confusing, and it is not actually used in the numerical calculations presented in the paper.
In particular, it is not clear what the author means when he states that he proposes the interpretation of the Hill-Wheeler equation as a quantum statistical probability. To the best of my understanding the Hill-Wheeler equation is already understood as a statistical probability for quantum tunneling processes over a barrier
If the author has developed a novel computational model with predictive capacities that simplify the current calculation of fission probabilities, the paper should focus on this model.
Reviewer 3 Report
Comments and Suggestions for Authors
The paper has improved.
What is needed now is to add a description of the Hill-Wheeler approach to statistical mechanics and to explain in such context, the origin of the formula to be used in the present manuscript.
Round 3
Reviewer 1 Report
Comments and Suggestions for Authors
Dear Editor,
I wish to recommend that this paper will be accepted for publication because it presents some intriguing observations and ideas, even though I still think these ideas are not consistently elaborated.
The paper notices the similarities between the energy spectrum of prompt neutrons from nuclear fission reactions and the spectrum of spontaneously emitted photons in semiconductors. This observation indeed suggests that this spectrum is related to a Fermi-Dirac distribution of nucleons (protons and neutrons) within the nucleus at some finite temperature and chemical potential.
It is also intriguing the analogy discussed between the authors between emitted photons in semiconductors and prompt neutrons in nuclei, even though such a claimed analogy should deserve a much more thorough discussion than the one presented by the author. For example, photons are massless bosons associated to a U(1) gauge symmetry, while neutrons are massive fermions with a positive baryonic charge.
The author goes on to try to use this observation to justify the identification of the Hill-Wheeler formula with the Fermi-Dirac distribution, which I really find meaningless. The Fermi-Dirac distribution describes the statistical occupation of single-particle fermionic states. In contrast, the Hill-Wheeler formula describes a probability for one of such fermions to quantum tunneling over a barrier.
The author presents interesting results describing collected experimental data for the fragments of fission processes and how they fit into the predictions of the Hill-Wheeler formula. Contrary to the author's claim, the shown agreement does not require interpreting the Hill-Wheeler formula as a Fermi-Dirac distribution.
As I said, even though I find this paper confusing, I'm willing to recommend that it be accepted for publication because it also contains interesting observations, and this is already the third round of revisions.
Reviewer 3 Report
Comments and Suggestions for Authors
The author's considerations revolve around what he calls the Hill-Wheeler formula
There is no such thing. The HW is a treatment, with many formulas.
This makes me doubt about the soundness of the work
I repeatedly asked that the author include into the paper a description of the HW approach, without success
I can not recommend acceptance at this stage